# Adversarial Robustness for Large Language NER models using Disentanglement and Word Attributions

**Xiaomeng Jin**[*†]   **Bhanukiran Vinzamuri**[‡]   **Sriram Venkatapathy**[‡]
**Heng Ji**[‡]   **Pradeep Natarajan**[‡]

[†]University of Illinois, Urbana Champaign  [‡]Amazon Alexa AI
xjin17@illinois.edu
{vinzamub, vesriram, jihj, natarap}@amazon.com

## Abstract

Large language models (LLM's) have been widely used for several applications such as question answering, text classification and clustering. While the preliminary results across the aforementioned tasks looks promising, recent work (Qin et al., 2023; Wang et al., 2023a) has dived deep into LLM's performing poorly for complex Named Entity Recognition (NER) tasks in comparison to fine-tuned pre-trained language models (PLM's). To enhance wider adoption of LLM's, our paper investigates the robustness of such LLM NER models and its instruction fine-tuned variants to adversarial attacks. In particular, we propose a novel attack which relies on disentanglement and word attribution techniques where the former aids in learning an embedding capturing both entity and non-entity influences separately, and the latter aids in identifying important words across both components. This is in stark contrast to most techniques which primarily leverage non-entity words for perturbations limiting the space being explored to synthesize effective adversarial examples. Adversarial training results based on our method improves the F1 score over original LLM NER model by **8**% and **18**% on *CoNLL-2003* and *Ontonotes 5.0* datasets respectively.

## 1 Introduction

Named Entity Recognition (NER) aims to identify and categorize named entities mentioned in unstructured text into pre-defined categories such as *Person*, *Location*, or *Organization*. In recent years, NER tasks (Malmasi et al., 2022) have become more challenging due to the introduction of complex tagsets, which often leads to the failure of existing NER systems in accurately recognizing these entities. To address this, Large Language Models (LLMs) have emerged as powerful tools, delivering significant performance improvements on NER tasks. However, these models tend to hallucinate with even minor modifications in the input (Wang et al., 2023b). Recently, there has been a growing interest in developing adversarial attack-based techniques for NER models (Simoncini and Spanakis, 2021; Lin et al., 2021) to enhance NER models using word-level attacks. This is even more relevant in the context of LLM's, as a very recent study (Zhu et al., 2023) demonstrated the lack of robustness in current LLM's with word-level attacks resulting in a significant performance drop of 33%. However, despite resulting in a successful attack, many of the perturbed word-level attack candidates fail to qualify as adversarial examples, i.e., they are not semantically similar to original sentences as depicted in Figure 1.

In addition, in resource constrained settings, it isn't frugal to explore the entire space of feasible word-level perturbations to devise good adversarial examples. This justifies the necessity of attack techniques which can explore the diverse space efficiently. In our paper, this is mainly accomplished using two key levers, namely, disentanglement and word attribution techniques, respectively. Disentanglement (Higgins et al., 2017) is a technique which helps in separating the latent entity and context components of an embedding space (Figure 2), making it more congenial for a word attribution function like Integrated Gradients (IG) (Sundararajan et al., 2017) to identify diverse, yet important words. The other subsequent steps include substitution of the selected words with entity and context substitution workflows (Figure 4), and selection of candidate adversarial examples based on their semantic similarity scores to the original text.

Experimental results indicate that to create a successful attack, on average, our method requires **69**% (details presented in Appendix A.1) lesser candidate adversarial sample generation than a state-of-the-art technique like CLARE (Li et al., 2021) when evaluated on three popular

---

[*]Work done while interning at Amazon Alexa AI

datasets (*CoNLL-2003* (Sang and De Meulder, 2003), *Ontonotes 5.0* (Weischedel et al., 2013), and *MultiCoNER* (Malmasi et al., 2022)). The results also demonstrate that our method improves the F1 score over original BERT NER model by **8**% and **18**% on *CoNLL-2003* and *Ontonotes 5.0* respectively. Furthermore, in an intrinsic evaluation of our adversarial examples generation approach on *CoNLL-2003*, we achieve a **10**% higher attack success rate (percentage of generated adversarial examples causing label flip) at a comparable modification rate. In addition, our method successfully attacks theinstruction fine-tuned T5 NER model (Wang et al., 2022) on the *MultiCoNER* dataset., resulting in a **10**% drop in the F1 score after the attack.

The main contributions in this paper are,

- We present a first of a kind architecture for synthesizing adversarial examples using a novel disentanglement technique and several components such as word attribution, word substitution and semantic similarity.
- Our novel disentanglement technique aids in generating a representation which significantly enhances the effectiveness of our attacks as outlined in our ablation studies.
- We present end-to-end results of improvements obtained through adversarial training on BERT, T5-based and LLAMA2 models using the examples generated from our approach on multiple datasets.

The rest of the paper is structured as follows. Section 2 provides an overview of related work in this field, highlighting the unique aspects of our contribution. In Section 3, we introduce our approach and discuss the individual technical components involved, including disentanglement, word attributions and semantic similarity. In Section 4, we present attack and adversarial training results on three benchmark datasets followed by conclusions.

## 2   Related Work

The existing adversarial attack methods on NER tasks can be classified into three categories: Character-level Attack, Word-level Attack, and Sentence-level Attack. The Character-level attacks generate adversarial examples by adding, deleting, or replacing a character in a word in the natural language texts. HotFlip (Ebrahimi et al., 2018) performs a character-level attack by swapping characters based on their gradient with respect to a

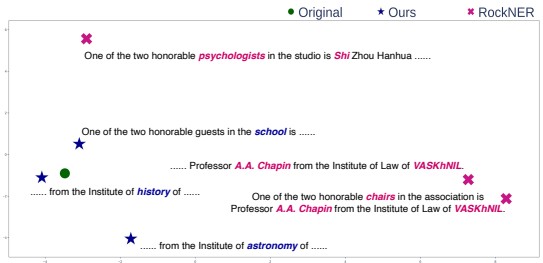

Figure 1: A PCA plot of an original sentence and the generated adversarial examples using different attack methods. The original sentence is *One of the two honorable guests in the studio is Professor Zhou Hanhua from the Institute of Law of the Chinese Academy of Social Sciences.*, represented with ●, the ★ are the adversarial examples from our method, and the ✖ markers represent the adversarial examples from RockNER. Across sentences we only retained modified spans and exclude identical content as compared to the original sentence.

one-hot input representation. However, these attacks have the problem that the character swapping generates spelling typos and the sentences are not semantically meaningful. Sentence-level attacks perform the operations by altering the input texts on the whole sentence. SCPN (Iyyer et al., 2018) includes a paraphrase generation under the guidance of a trained parser to label syntactic transformation.

Word-level attacks are more popular and effective methods than the above-mentioned two methods. There are also many effective word-level attacks on text classification tasks (Ribeiro et al., 2020; Das and Paik, 2022). (Liu et al., 2021) proposes an efficient local search algorithm to determine the possible word substitutions. However, the adversarial attacks on sequence-to-sequence models are not much explored. (Simoncini and Spanakis, 2021) extends TextAttack (Morris et al., 2020) framework that consists of multiple attack strategies via reformulating the goal functions to support NER tasks. RockNER (Lin et al., 2021) is a simple NER adversarial attack by perturbing both named entities and contexts in the original texts. However, the generated adversarial examples all have the issue of poor semantic equivalence to the original input sentences because of the high modification rate. In contrast, we propose a word-level adversarial attack on NER models that effectively identifies important words via word attributions and generates the adversarial examples with a lower modification rate.

Recently, large language models (LLM's) (Brown et al., 2020; Wang and Komatsuzaki, 2021;

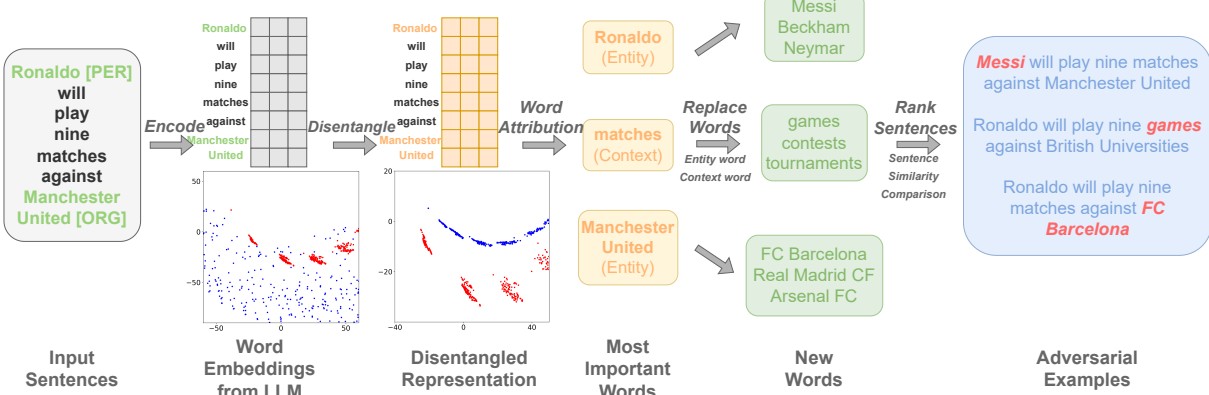

Figure 2: The architecture of our method as depicted here proceeds as follows. Given an input text, we first obtain word embeddings from language models that the NER model uses, then learn a disentangled representation for entity and context representations (context, entity in the UMAP embedding scatter plots). The disentangled representations are then used to compute scores via the IG word attribution function. The most important words are replaced with new words following word substitution workflows. After generating new sentences, we rank the sentence similarity scores and output the most similar sentences as adversarial examples.

Touvron et al., 2023), have brought attention that they outperform many NLP tasks and have been widely used for applications (Sallam, 2023). Nori et al. (2023) explore potential use of LLMs in medical education, assessment, and clinical practice. However, LLMs still underperform in some NLP tasks. (Wang et al., 2023a) study the robustness of in-context learning and propose an ICL attack on large language models by manipulating the demonstrations. In this paper, we focus on instruction fine-tuned language model on NER tasks and examine its robustness under our proposed adversarial attack method.

Disentanglement (Higgins et al., 2017) has been primarily used in the space of auto-encoders for learning latent factors which are mutually orthogonal (diverse) w.r.t one another to aid with interpretability primarily for image-based applications. They have also been extended for text-based tasks (Zou et al., 2022) to leverage the latent components to learn more desired downstream representations. In this paper, we use disentanglement to segregate the entity and non-entity representations in the feature space.

## 3 Our Method

In this section, we describe the threat model and the key technical components of our proposed adversarial NER framework. For the sake of brevity, we refer to *non-entity* as *context* and *LLM NER* as NER throughout the rest of this paper. The approach consists of the following steps, (1) Disentanglement of word representations, (2) Selection of important words using word attributions, (3) Substitution of the selected words with candidate alternatives, (4) Ranking and selection of candidate adversarial examples

### 3.1 Threat Model

The threat model is composed of two key components the adversary and the defender

#### 3.1.1 Goals of the adversary

The adversary's primary goal is to manipulate the NER model's predictions by introducing subtle changes to the input text that cause the model to misclassify named entities.

#### 3.1.2 Capabilities of the adversary

The adversary has a deep understanding of the NER model's architecture and training data. The adversary has access to the disentanglement and word attribution modules to sample entity and or non-entity component words. In addition, the adversary has access to two different word substitution workflows for entities and context, respectively. Description of these modules are provided in the following subsections within this section.

#### 3.1.3 Knowledge of the adversary

The adversary has knowledge of the NER model's training data and can use this knowledge to craft adversarial examples that are specifically designed to cause mis-classifications through LLM hallucination and/or any other mechanisms.

### 3.1.4 Goals of the defender

The defender's primary goal is to protect the NER model from adversarial attacks and ensure that it continues to make accurate predictions.

### 3.1.5 Capabilities of the defender

The defender has access to techniques for adversarial training, which can help to increase the NER model's robustness to attacks.

### 3.1.6 Knowledge of the defender

The defender has knowledge of the NER model's architecture and training data, as well as any potential vulnerabilities that may be exploited by the adversary. The defender may also have knowledge of common adversarial attack strategies such as Bert-Attack (Li et al., 2020), DeepWordBug (Gao et al., 2018), etc.

### 3.2 Disentanglement of Word Representations

The primary objective of disentanglement is to mitigate bias (between entity and non-entity words) in word selection for perturbation, thereby increasing the diversity of the generated adversarial examples. In Table 1, we provide supporting evidence that indicates that NER models are highly biased towards context words during prediction. The word attribution function like IG picks 98.9% of non-entity words from the CoNLL dataset for generating potential adversarial examples if disentanglement is not applied. This adversely affects the overall diversity of the generated adversarial examples thereby preventing exploration of potential vulnerabilities of the NER model to entity specific perturbations. Our approach of disentanglement mitigates this problem through generating adversarial examples by exploring the entire perturbation space across important entity and non-entity words in an efficient manner.

Given the original features of an input sentence, we first split word features into two sets: $E$ for features of entity words and $C$ for features of context words. Our goal is to learn a new representation $\hat{E}$ for entity words, so that the features of entity words are independent of context words. The resulting representation $(\hat{E}, C)$ will be utilized for calculating word attribution scores in the next step.

Following (Marx et al., 2019), we learn the disentangled representations by using an auto-encoder. The architecture of the auto-encoder is illustrated in Figure 3, which consists of three neural networks: the encoder, the decoder, and the discriminator:

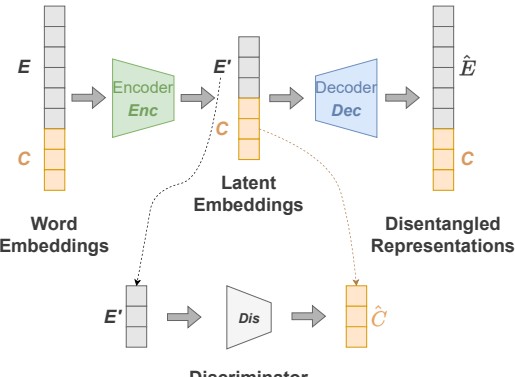

Figure 3: The architecture of disentangled representation learning framework which takes initial word feature embeddings as inputs, and use an auto-encoder to learn a latent representation $E'$ for entity words. A discriminator $Dis$ is used to recover context embeddings $C$ from entity embeddings $E'$

**Encoder**. The encoder $Enc$ takes entity word embeddings $E$ as input, then learn a disentangled representation $E'$ in the latent space. Note that the encoder is also aware of context word embeddings $C$ when encoding $E$:

$$Enc(E; C) = E'. \qquad (1)$$

**Decoder.** The decoder $Dec$ takes $E'$ as input and output $\hat{E}$, which is a new (disentangled) representation for $E$:

$$Dec(E'; C) = \hat{E}. \qquad (2)$$

**Discriminator.** The discriminator $Dis$ tries to predict (recover) the context embeddings $C$ using the hidden representation of entity words $E'$:

$$Dis(E') = \hat{C}, \qquad (3)$$

where $\hat{C}$ is the predicted result for $C$. However, since we expect $E'$ and $C$ to be independent, the recovery error between $\hat{C}$ and $C$ should be maximized. In other words, the discriminator cannot recover any information of $C$ using $E'$, which indicates that there is no correlation between $C$ and $E'$ (as well as $\hat{E}$).

**Training objective.** Our training process aims to minimize the reconstruction mean squared error (MSE) between $E$ and $\hat{E}$, as well as maximizing the recovery error between $C$ and $\hat{C}$:

$$L = \mathrm{MSE}(E, \hat{E}) - \beta \cdot \mathrm{MSE}(C, \hat{C}), \qquad (4)$$

| Dataset | CoNLL2003 | OntoNotes |
|---|---|---|
| **Original Distribution of Context** | 83.2% | 91.1% |
| **Distribution of Context as most important words before disentanglement** | 98.9% | 96.1% |
| **Distribution of Context as most important words after disentanglement** | 81.3% | 91.4% |

Table 1: Word type distribution of two datasets. The first row represents the original distribution of with the NER label O (context words) among all word labels in the two datasets followed by two rows representing the distribution of important context words identified by the IG word attribution function in two different settings. It is clear that in comparison to before disentanglement (the second row), the representation learned using our balancing disentanglement technique (in third row) obtains a distribution that aligns more closely with the original dataset distribution (first row).

where $\beta$ is a balancing hyper-parameter. $\hat{E}$ and $C$ are taken as the disentangled representations for entity and context words.

### 3.3 Selection of Important Words

To evaluate the impact of each word's feature on the NER model's output, we propose to compute the word attribution scores (a.k.a feature importances or saliency scores) using Integrated Gradients (IG) (Sundararajan et al., 2017). IG calculates the gradient of the model's prediction output with respect to its input features and returns the attributions of output labels with respect to the input features. The assigned value on each input feature is the importance score to model outputs. The mathematical formulation for IG is as follows

Suppose we have a black-box machine learning model $f$, an input $x \in R^n$, and a baseline input $x' \in R^n$ (for text models it could be a zero embedding vector). The Integrated Gradient (IG) along the $i^{th}$ dimension for the input $x$ and $x'$ is defined as:

$$IG_i(x) ::= (x_i - x_i') \times \int_{\alpha=0}^{1} \frac{\partial F(x' + \alpha(x - x'))}{\partial x_i} d\alpha. \tag{5}$$

We obtain a ranked list of words in the sentence by sorting them based on their word attribution scores. The top-K among them are then selected for perturbations. The use of disentangled word representations (as described in the earlier section) enables a more balanced selection.

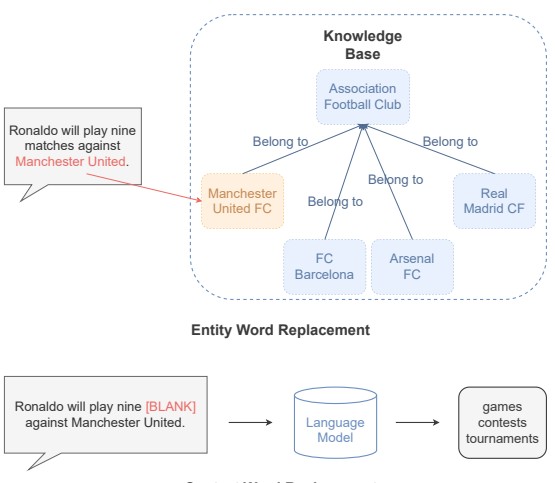

Figure 4: Entity and Context Word Substitution Workflows

### 3.4 Word Substitution of Selected Words

After identifying the most important words in a sentence, the next step is to determine potential substitutions for these words. We outline both replacement workflows in Figure 4 and briefly describe them below.

**Named Entity Substitution** To keep the entity labels unchanged, the replacement word for a named entity word should belong to the same entity type as the original entity word. Since Wikidata contains abundant structured knowledge and most of the entities have corresponding entries in it, we utilize it as as an external knowledge base for word replacement. Specifically, similar to (Lin et al., 2021), to determine the substitution for a given named entity, we first use entity linking tools (Honnibal et al., 2020) to link the named entity to an entry in Wikidata. If a matching entry is found in Wikidata, we collect the entries that have a *belonged to* relation with this entry and take them as the upper categories of the target named entity. We randomly sample at most 10 of the entries under the same upper categories as possible word substitutions.

As an example illustrated in Figure 4, the label of the target word **Manchester United** is an organization. After linking this entity to the corresponding entry *Manchester United FC* in the Wikidata knowledge base, we find that *Manchester United FC* belongs to the category of *Association Football Club*. Within the same category, we find other entries such as *FC Barcelona*, *Real Madrid CF*, and *Arsenal FC*. These entities are possible substitutions to replace the original entity word **Manch-**

**ester United**.

**Context Word Substitution** We leverage an in-filling language modeling framework (ILM) (Don-ahue et al., 2020) that allows language models fill in the blanks given masked texts. We use a large language model (LLM) GPT-J (Wang and Komat-suzaki, 2021) to generate texts. Specifically, we mask the context word that needs to be replaced with [BLANK]. Then, we take the masked sentence as input to the LLM with a prompt that asks the model to generate 5 possible candidate words to substitute the [BLANK] word. Finally, we collect the output words as substitutions for the context word, as illustrated in Figure 4.

### 3.5 Sentence Ranking

For an input sentence, we identify target words to be substituted and replace them with new words, which gives us a set of new sentences. To decide which sentences should be output as the final adversarial samples, we compute the similarity between each new sentence and the original one using Universal Sentence Encoder (USE) (Cer et al., 2018). It encodes natural language texts into high dimensional vectors as text embedding. After encoding each pair of original and generated sentences, we compute their cosine similarity as their similarity score. Basically, given the original and new sentences $S_1$ and $S_2$, the similarity score is computed by:

$$Sim(S_1, S_2) = \frac{USE(S_1) \cdot USE(S_2)}{\|USE(S_1)\| \, \|USE(S_2)\|} \quad (6)$$

## 4 Experiments

### 4.1 Datasets

We conduct our experiments on three datasets, *CoNLL-2003* (Sang and De Meulder, 2003), *Ontonotes 5.0* (Weischedel et al., 2013), and Mul-tiCoNER (Malmasi et al., 2022). *CoNLL-2003* is a NER dataset that consists of named entities classified into four types: persons, locations, organizations, and names of miscellaneous entities that do not belong to the previous three groups. *Ontonotes 5.0* is a large corpus of news articles in three languages, annotated with 18 types of named entities. MultiCoNER is a complex NER dataset that covers 11 languages and has a fine-grained tagset with 36 entity types. In our experiments, we focus on the English track of these datasets.

| Dataset | CoNLL | OntoNotes | MultiCoNER |
|---------|-------|-----------|------------|
| **Train** | $14,987$ | $59,924$ | $16778$ |
| **Val** | $3,466$ | $8,528$ | $871$ |
| **Test** | $3,684$ | $8,262$ | $871$ |

Table 2: Train, Validation and Test Splits.

### 4.2 Baseline Methods

We compare our proposed method with the following NER adversarial attack baseline methods:

- **RockNER** (Lin et al., 2021) creates adversarial examples by operating at the entity level and replacing all target entities with other entities of the same semantic class in Wikidata. At the context level, it randomly masks up to 3 context words and uses pre-trained language models (PLM) to generate word substitutions.
- **SeqAttack** (Simoncini and Spanakis, 2021) is an attack framework against token classification models. The framework extends TextAttack (Morris et al., 2020) and contains multiple adversarial attack strategies. We compare our attack method with 3 attacks in the framework, including Bert-Attack, CLARE, and DeepWordBug.

### 4.3 Experimental Setup

For the choice of the NER model, we follow the same settings as the baseline methods for fair comparison. We utilize three different models for our experiments: a BERT base model cased (Lafferty et al., 2001; Devlin et al., 2019), a T5 language model (Raffel et al., 2020) (T5-LARGE 770 M parameter model) and LLAMA 2-7B-CHAT model (Touvron et al., 2023). Architectures for BERT and T5 NER models are provided in the Appendix in Figure 7 and Figure 8, respectively.

For the BERT NER model, we perform fine-tuning on both the *CoNLL-2003* and *OntoNotes* datasets. Regarding the T5 model, we transform the NER task into an instruction fine-tuned model (T5 Instruction NER(Wang et al., 2022)) as outlined in the Appendix in A.3. This is facilitated by converting the *MultiCoNER* dataset, into sentence-instruction pairs, we follow a similar approach to the sentences illustrated in Figure 6 in the Appendix.

To compute the word attributions, we freeze the parameters of encoder of the T5 model after fine-tuning, and add an additional classification layer after it, to create a new sequence labeling model. This T5 encoder only model is trained with a CRF

loss, and we obtain word attributions and generate adversarial examples by directly utilizing the predictions of this model. We evaluate the effectiveness of our attack methods based on multiple evaluation metrics as outlined below:

### 4.3.1 Evaluation Metrics

- **Attack Rate**. This metric measures the success of an attack on a sentence. An attack is successful on a sentence if the NER model misclassifies at least one named entity in the sentence after the attack. We calculate the rate of successful attacks among all attacks.
- **Modification Rate**. The modification rate evaluates the changes made to the original sentence during the attack. It is calculated as the percentage of different tokens between the original sentence and the modified sentence.
- **Textual Similarity**. Textual similarity is computed between the original sentence and the generated sentence. We use Universal Sentence Encoder (USE) to encode each sentence and compute cosine similarity between the two vectors.

### 4.3.2 Attack and Hyper-parameter Settings

As shown in Table 3, to test the attack performance of our method, we report results at different levels of modification rate (maximum 1 word or 3 words). All displayed results are averaged across 3 random runs.

For the BERT NER model, we perform fine-tuning on a pre-trained bert-base-cased model with CRF. The training process utilizes a batch size of 32 and a learning rate of 1e-5. We conduct training for a total of 20 epochs on both the CoNLL-2003 and OntoNotes datasets. In the case of the T5 model, we conduct a fine-tuning on the *Multi-CoNER* dataset. The training process has a batch size of 16 and a learning rate of 3e-4. The total training epoch is set to 10.

The encoder, decoder and discriminator for computing the disentangled representations consist of two hidden layers with size = [64, 10] and $\beta$ is set to 0.5. For IG word importance computation, we leverage the Captum library from Meta (Kokhlikyan et al., 2020). The learning rate is 1e-3, and the number of epochs is 25. We use Stochastic Gradient Descent optimizer with weight decay 1e-5.

After computing the word importance, we take the top K = 1, 3 important words to replace. Then,

we replace the most important words as in Section 3.4 to output adversarial examples.

### 4.4 Benchmarking Results

In this section, we primarily benchmark the BERT NER, T5 Instruction NER and LLAMA 2 models across different attack strategies.

### 4.4.1 Comparison against baselines

In Table 3, for the *CoNLL-2003* dataset, we present a comparison of the performance of our method against the five adversarial attack strategies within the SeqAttack framework. To ensure a fair comparison, we fine-tune our model to achieve the same F1 score as the original SeqAttack results (98%) before conducting the attack.

From the results in Table 3, we observe that our method shows a comparable level of token modification rate (at around 22%) to both Bert-Attack and DeepWordBug-II$_{30}$. However, our method outperforms Bert-Attack by achieving approximately 10% higher attack success rate. Furthermore, when compared to DeepWordBug-II$_{30}$, our method achieves a lower F1 score by 8% and a higher attack success rate by 24%. These results show the effectiveness of our method in achieving successful attacks with a lower level of modification rate.

These significant improvements in F1 score and attack rate while having similar modification rates are attributed to the following reasons, a) our unique approach of combining effective disentangled representation learning with IG-based targeted word attributions, and b) using semantic similarity based guardrails while generating adversarial examples.

In Table 3, for the *OntoNotes* dataset, we compare the performance of RockNER and our method. For both RockNER and our method, successful attacks would result in lowering the F1 score compared to the baseline F1 score.

Note that RockNER chooses semantic-rich words and randomly masks tokens (3 max) in context-only attack, and replaces all named entities in entity-only attack. Different from RockNER, we compute disentangled representations and replace the most important 1 or 3 words in the sentences.

Comparing the F1 scores under attack, our method is much more effective than the baseline method on all three types of attacks. More importantly, we observe that the F1 score for our method is 4% lower than the RockNER. This demonstrates that our method is better at selecting and replac-

| Dataset | Attack Name | F1 Score↓ | Attack Rate↑ | Mod Rate↓ | Text Sim↑ |
|---|---|---|---|---|---|
| CoNLL | **Bert-Attack** | 79% | 44% | 22% | 84% |
| | **CLARE** | 79% | 37% | 70% | 86% |
| | **DeepWordBug-II**$_{30}$ | 87% | 30% | 21% | 83% |
| | **RockNER** | 80% | 54% | 35% | 64% |
| | | 82% | 36% | 11% (max 1 word) | 91% |
| | **Our Method** | **79%** | **54%** | 22% (max 3 words) | 84% |
| | **Original** | 98% | - | - | - |
| OntoNotes | **RockNER** | 55% | 37% | 26% | 66% |
| | **Our Method** | 65% | 33% | 6% (max 1 word) | 86% |
| | | **51%** | **42%** | 12% (max 3 words) | 79% |
| | **Original** | 90.3% | - | - | - |

Table 3: Results using different attack strategies on BERT NER model for *CoNLL-2003* and *OntoNotes* datasets. The first 3 attack methods are derived from SeqAttack. Numbers in the 4th and the 8th rows are derived using RockNER baseline. We generate adversarial examples allowing for a maximum of 1 or 3 word replacements. *Lower* F1 score and modification rate along with *higher* attack rate and textual similarity indicate a more superior attack strategy as demonstrated by our method.

| Method | Attack Rate↑ | Mod Rate↓ |
|---|---|---|
| **RockNER** | **26.7%** | 40.2% |
| **Random** | 18.2% | 19.7% |
| **Our Method** | 25.6% | **19.7%** |

Table 4: Attack Rate and Modification Rate for LLAMA2-7B-CHAT model on the CoNLL-2003 dataset.

ing the most important words which enables it to generate more effective adversarial examples.

### 4.4.2 Attack Results with LLAMA 2 model

To examine the effectiveness of our attack method, we conduct experiments using LLAMA2-7B-CHAT model (Touvron et al., 2023) on the CoNLL-2003 dataset. For this experiment, we allow a maximum of 1 word perturbation following our proposed framework. We compare our results with two baseline methods, RockNER and Random replacement of a single word. Comparing the results in Table 4, we observe that both our proposed method and RockNER achieve a higher attack success rate than random replacement. In addition, our method has a much lower modification rate compared to RockNER. After adversarial training, we observe that the robustness improves and F1 under attack increases by 3.1%.

### 4.4.3 Attack Results with T5 Instruction NER model

For the *MultiCoNER* dataset, the clean F1 score on the test set is 47.23% with the T5 Instruction NER model. We follow the same attack framework as

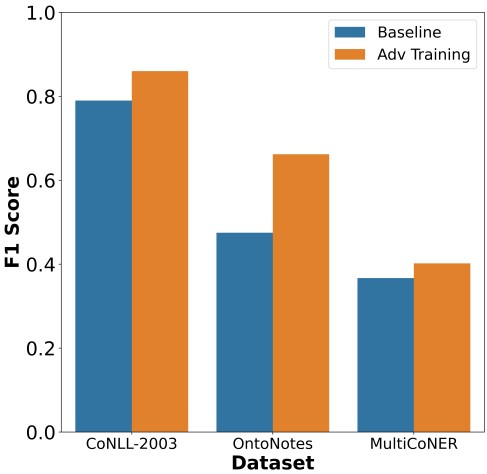

Figure 5: **F1 score**↑ results on three datasets. Blue columns and orange columns show the F1 score under attack before and after adversarial training, respectively.

described above and allow max 1 word perturbation, and generate adversarial examples from the original test set. On the new adversarial test set, the F1 score under attack drops to 37.96% with T5 instruction NER model. To better demonstrate the effectiveness of our method, we also use the baseline method RockNER to generate the adversarial test set, which replaces all named entities and allows max 3 word context words perturbation. The F1 score under attack of T5 instruction NER model is 36.87% for RockNER. Our method reaches a comparable attack performance, whereas our mod-

| Disentanglement | | F1 Score↓ | Attack Rate↑ |
|---|---|---|---|
| CoNLL | Original | 98% | - |
| | without | 86% | 25% |
| | with | **82%** | **36%** |
| OntoNotes | Original | 98% | - |
| | without | 69% | 28% |
| | with | **51%** | **42%** |

Table 5: Attack Results for BERT NER model with and without disentanglement steps.

| Word Selection | CoNLL | OntoNotes |
|---|---|---|
| Original | 98% | 90% |
| Random | 87% | 61% |
| TF-IDF | 90% | 58% |
| Kernel-SHAP | 86% | 61% |
| Integrated Gradients (IG) | **82%** | **51%** |

Table 6: F1 scores for BERT NER model under attack using different strategies. (*lower is better*)

ification rate is only 46% of the modification rate from RockNER. Further details on the setup here are provided in the Appendix in Section A.3.

### 4.4.4 Ablation Studies

**Without disentanglement** To assess the effectiveness of our disentanglement technique, we compare the attack performance on the BERT NER model with and without disentanglement steps. The results, as illustrated in Table 5, demonstrate the impact of disentanglement on the attack success rate. With the disentanglement step, we observe a reduction in bias in word selection (Refer Table 1) improving diversity of adversarial samples generated and a significant attack performance improvement. **Other word selection methods** To examine the effectiveness of IG, we also tested other word selection strategies. (1) Randomly select words for substitution. (2) Select words using TF-IDF values as word importance. (3) Select words using KernelSHAP (Lundberg and Lee, 2017) that uses a special weighted linear regression to compute feature importance. Compared with F1 scores in Table 6, IG outperforms all other three word selection methods, which proves the effectiveness of using IG. The results that IG outperforms other three methods in non-linear models are also consistent with insights here (Modarres et al., 2018).

### 4.4.5 Adversarial Training

After generating adversarial examples from the training set using the workflow depicted in Figure 2, we conduct adversarial training on the trained NER models on the three datasets. As illustrated in Fig-

ure 5, we observe that adversarial training enhances model robustness across all datasets. Specifically, for adversarial examples that allow maximum 3 word replacement on *CoNLL-2003* dataset, the F1 score under attack improves by approximately **8%** after adversarial training. Similarly, the F1 score under attack on the *OntoNotes* dataset demonstrates an improvement of about **18%** after adversarial training. In the case of the *MultiCoNER* dataset, we observe a 4% improvement in the F1 score under attack after adversarial training.

## 5 Conclusions

LLM's for complex NER problems have been gaining a lot of traction recently, and in this work, we propose an adversarial attack based framework to make these NER models more robust to widen their adoption even further. Our end-to-end approach combines a novel disentanglement technique with word attributions, substitution and semantic similarity to generate adversarial examples. Our disentanglement method builds upon the idea of trying to learn an embedding by disentangling entity and non-entity latent representations

Applying disentanglement before computing IG word attributions aids in ensuring that we are able to synthesize a diverse set of adversarial examples in an extremely efficient manner as demonstrated through lower candidate sample generation and modification rates. Experimental results across BERT, T5-based and LLAMA 2 NER models on three benchmark datasets demonstrates that our method significantly outperforms competing baseline methods. Ablation studies highlight the importance of disentanglement and word attribution techniques.

## 6 Limitations

One of the key limitations of our work is that we have not explored the entire LLM landscape and would be keen to explore the decoder only models. Being able to evaluate our work across decoder only models strengthens some of the key claims made in this paper. In particular, we would like to investigate robustness of models with growing size.

Future avenues to investigate for us mainly include building an end-to-end framework to synthesize attacks for LLM NER models where the word attributions and semantic similarity scoring are pursued as potential paths within a pipeline to identify

the optimal attack. This will help in reducing the dependence on specific choices significantly.

## Ethical Consideration

We acknowledge that our work is aligned with the *ACL Code of the Ethics* [1] and will not raise ethical concerns. We do not use sensitive datasets/models that may cause any potential issues/risks.

## Acknowledgement

We thank the anonymous reviewers for their helpful suggestions. This research work is supported by Amazon Alexa AI, the Molecule Maker Lab Institute: an AI research institute program supported by NSF under award No. 2019897 and No. 2034562, and DOE Center for Advanced Bioenergy and Bioproducts Innovation (U.S. Department of Energy, Office of Science, Office of Biological and Environmental Research under Award Number DESC0018420). The views and conclusions contained herein are those of the authors and should not be interpreted as necessarily representing the official policies, either expressed or implied, of the Molecule Maker Lab Institute, or the U.S. Department of Energy. The U.S. Government is authorized to reproduce and distribute reprints for governmental purposes notwithstanding any copyright annotation therein.

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

## A  Appendix

### A.1  Model Queries Comparison

We compare our attack performance with CLARE under SeqAttack framework. CLARE is a word-level attack technique which generates highest-scoring candidate sentences from replacing, inserting, and merging new words into the original sentences. According to their experimental results in the paper, it reaches F1 score 79% and attack success rate 37% whereas our attack method gets F1 score 79% and attack success rate 54%. CLARE attack allows at most **512** model queries. It generates all possible new sentences (usually 1000+) and checks if each new sentence attacks the NER model successfully. The attack success rate is very low. Among the successful attack sentences, it takes on average about **33** model queries (33 new sentences) to reach a successful attack. Compared with our method, we only allow at most **10** candidate sentences generated from each original sentence. Then we check if any sentence is a successful adversarial example which is much less than the baseline method.

Some example output from CLARE:
**Attacking sample**:  Japan began the defence of their Asian Cup title with a lucky 2-1 win against Syria in a Group C championship match on Friday .

**AttackedText**: "Japan 's began the defence of their Asian Cup title with a lucky 2-1 win against Syria in a Group C championship match on Friday ."

**AttackedText**: "Japan  began the defence of their Asian Cup title with a lucky 2-1 win against Syria in a Group C championship match on Friday ."

**AttackedText** "Japan A began the defence of their Asian Cup title with a lucky 2-1 win against Syria in a Group C championship match on Friday ."

**AttackedText**: "Japan a began the defence of their Asian Cup title with a lucky 2-1 win against Syria in a Group C championship match on Friday ."

**AttackedText**: "Japan ai began the defence of their Asian Cup title with a lucky 2-1 win against Syria in a Group C championship match on Friday ."

**AttackedText**: "Japan ain began the defence of their Asian Cup title with a lucky 2-1 win against Syria in a Group C championship match on Friday ."

**AttackedText** "Japan ama began the defence of their Asian Cup title with a lucky 2-1 win against Syria in a Group C championship match on Friday ."

**AttackedText** "Japan an began the defence of their Asian Cup title with a lucky 2-1 win against Syria in a Group C championship match on Friday ."

**AttackedText** "Japan and began the defence of their Asian Cup title with a lucky 2-1 win against Syria in a Group C championship match on Friday ."

**AttackedText** "Japan ans began the defence of their Asian Cup title with a lucky 2-1 win against Syria in a Group C championship match on Friday ." (All generated examples failed to attack)

From the outputs using CLARE, we can see that their attack strategy is to select multiple indices in the sentence for word-level operations. In the attacking sample, the attack method iteratively inserting random words between *Japan* and *began*. These random insertion incurs syntax errors in the generated sentences. Comparing with this baseline, our method utilizes word attribution to find the most important word in the sentence more efficiently. In addition, with the help of PLMs and knowledge bases, we are able to replace with more reasonable words.

### A.2  F1 score under attack results on OntoNotes dataset

In Table 8, for the *OntoNotes* dataset, we compare the performance of RockNER and our method under three different attack settings, namely, a) context-only attacks, b) entity-only attacks and c) context + entity attacks. The second column in this table represents the baseline F1 scores on the original test set before the attack was conducted. For both RockNER and our method, successful attacks would result in lowering the F1 score compared to the baseline F1 score.

Note that RockNER chooses semantic-rich words and randomly masks tokens (3 max) in context-only attack, and replaces all named entities in entity-only attack. Different from RockNER, we compute disentangled representations and replace the most important context words (3 max) for context-only attack and the most important entity words (3 max) for entity-only attack.

Comparing the F1 scores under attack, our

| Original Test Sentence | Adv Example (RockNER) | Adv Example (Our Method) |
|---|---|---|
| **Sentence:** Dear viewers, the China News program will end here. **Output:** China News is an organization. | **Sentence:** Dear viewers, the *Hiwwe wie Driwwe people* will *lose* here. **Output:** Hiwwe wie Driwwe is not an entity. | **Sentence:** Dear viewers, *my* China News program will end here. **Output:** China News is a work-of-art. |
| **Sentence:** Relevant departments from Beijing Municipality promptly activated emergency contingency plans. **Output:** Beijing Municipality is a geopolitical entity. | **Sentence:** *Related* departments from Markham promptly activated emergency contingency *members.* **Output:** Markham is not an entity. | **Sentence:** Relevant departments from *Berlin* Municipality promptly activated emergency contingency plans. **Output:** Berlin Municipality is an organization. |

Table 7: A case study on generated adversarial examples by our method. The sentences on the left are original test sentences from *OntoNotes* dataset with correctly predicted named entities (blue colored words). The sentences in the middle and on the right are generated adversarial examples using RockNER baseline and our method that cause the NER model make wrong predictions. All input sentences also come with the instruction *Please extract entities and their types from the input sentence.* The *italic* words are the candidate words got selected and replaced. The red labels are incorrect predictions after the adversarial attacks).

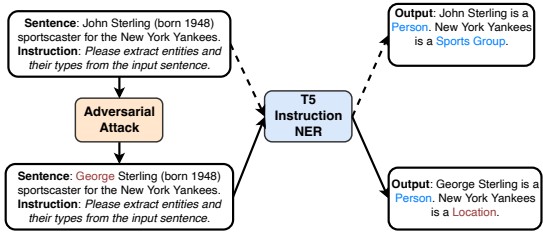

Figure 6: Input sentence from *MultiCoNER* dataset and the adversarial example generated using our attack method, wherein we attack a instruction fine-tuned T5 NER model here. Following our attack, the NER model incorrectly classifies *New York Yankees* as a *Location*.

method is much more effective than the baseline method on all three types of attacks. More importantly, we observe that the F1 score for our method is $6\% \sim 8\%$ lower than the RockNER. This demonstrates that our method is better at selecting and replacing the most important words which enables it to generate more effective adversarial examples.

### A.3 Adversarial Examples for Instruction Fine-tuned Model

We fine-tune a T5 language model for named entity recognition task on MultiCoNER dataset. As shown in Figure 8 on the left, T5 is an encoder-decoder model and we convert the NER dataset into a text-to-text format. To compute the influence score, after fine-tuning the T5 model, as shown in Figure 8 on the right, we take its encoder only, freeze the parameters, and add one more linear layer after it to form a new sequence labeling model. Then, we train the last linear layer with CRF loss.

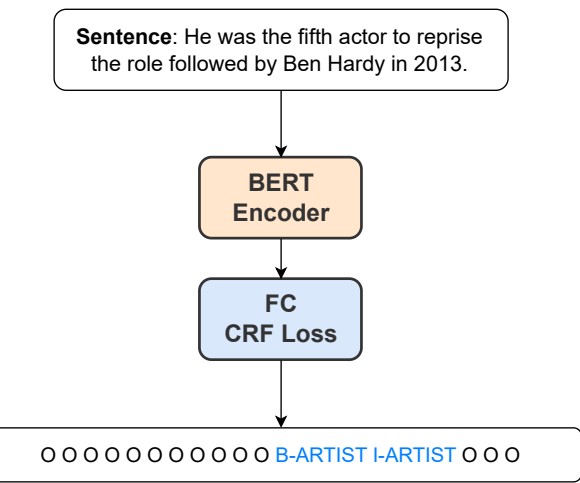

Figure 7: The architecture of the BERT-CRF NER model.

In this way, we can directly take the output logits of this T5-encoder only model for word attribution computation.

Figure 6 presents an example sentence generated through an adversarial attack that successfully perturbs the NER model. The sentence on the top comes from the original training set. Along with the instruction *Please extract entities and their types from the input sentence*, the T5 Instruction NER model correctly predicts the NER labels in natural language texts format.

We propose an approach for word-level adversarial attacks that generates adversarial examples in a principle manner. In the example presented in Figure 6, the word 'John Sterling' is an entity word that influences the NER predictions. Then,

| Model Name | F1 Score Original test | F1 Score | | |
|---|---|---|---|---|
| | | Context-only ↓ | Entity-only ↓ | Context + Entity ↓ |
| **BERT-CRF (RockNER)** | 90.6% | 85.8% | 59.2% | 54.6% |
| **BERT-CRF (Ours)** | 90.3% | 79.6% | 51.1% | 47.5% |

Table 8: F1 score under attack results on *OntoNotes* dataset. Both RockNER and our method are evaluated under context-only attack, entity only attack, and context+entity attack settings. Comparing numbers row-wise against RockNER (first row) reveals that our method (second row) obtains *lower* F1 score compared to F1 score on original test set under all three attack settings.

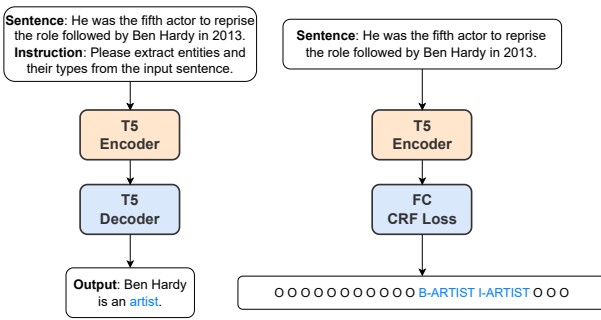

Figure 8: The architecture of two T5 models. On the left is the T5 Instruction NER Model and on the right is the T5 Vanilla NER Model.

method are more semantically similar to the original test sentence. This can be attributed to the fact that our method synthesizes diverse and effective adversarial examples while preserving textual similarity. (keeping the modification rate low).

we replace it with relevant substitutes to generate the adversarial examples. The disentanglement of the latent representations of entity or non-entity word types prevents bias towards perturbations of a particular word type.

## A.4 Case Studies

We show the generated examples from *OntoNotes* dataset using the RockNER baseline method and our method in Table 7. The sentences on the left are from the original test set and the blue texts are correctly predicted named entities by the NER model. The sentences in the middle are the generated adversarial examples using the RockNER baseline method. The sentences on the right are the generated adversarial examples following the workflow in Figure 2. The *italic* words are selected to be replaced from the original sentences. The red labels are wrong predictions by the NER model after the adversarial attacks.

In this experiment, we allow at most one word replacement and it could either be context or a named entity word. From the output we can see that, using our method, the replacement of one word effectively causes the NER model to make wrong prediction.

From Table 7, we observe that in comparison to RockNER, the adversarial examples from our