# OpenReview forum: "Adversarial Robustness for Large Language NER models using Disentanglement and Word Attributions"
_EMNLP/2023/Conference — EMNLP 2023 Findings_

### Official Review · Reviewer_MbYs · 2023-07-26

**Typos Grammar Style And Presentation Improvements:** NA
**Soundness:** 4

**Excitement:**

4: Strong: This paper deepens the understanding of some phenomenon or lowers the barriers to an existing research direction.

**Missing References:**

NA

**Paper Topic And Main Contributions:**

This paper proposes a novel adversarial attack for NER models. The attack consists of disentanglement and word attribution techniques to address resource-constrained attacks. Experiments show the effectiveness of the proposed methods.

**Questions For The Authors:**

NA

**Reasons To Accept:**

* The paper is well-written and easy to follow.

* The proposed method, consisting of disentanglement and word attribution, is novel.

**Reasons To Reject:**

The author frequently highlights LLM, but their experiments are not compared with LLM models such as GPT-3.5 or Llama.

**Reproducibility:**

4: Could mostly reproduce the results, but there may be some variation because of sample variance or minor variations in their interpretation of the protocol or method.

**Reviewer Confidence:**

2: Willing to defend my evaluation, but it is fairly likely that I missed some details, didn't understand some central points, or can't be sure about the novelty of the work.

---

> ### Author Rebuttal · Authors · 2023-08-29
>
> We thank the reviewer for their valuable inputs and feedback. We appreciate the reviewer recognizing the novelty and effectiveness of our proposed method. Our response to your questions are as follows:
>
> **[Comparison with Llama]:** We now provide a comparison with Llama2 also here.
>
> | Adversarial Attack Results      | Attack Rate | Modification Rate |
> | ----------- | ----------- | ----------- |
> | Our Method    |  25.6%   |   19.7%   |
> | Baseline (RockNER)   |  26.7%      |  40.2%    |
> | Random  |  18.2%      |   19.7%    |
>
> In the table above, we report results using the Llama2-7b-chat-hf model on the CoNLL-2003 dataset. For this experiment, we allow a maximum of one word perturbation following our proposed framework. The first baseline method that we are comparing is RockNER, which by default substitutes all named entities as well as a maximum of three randomly selected context words. The second baseline method is the random replacement of a single word. Comparing the results of all three methods shown in the table, we observe that both our proposed method and RockNER achieve a higher attack rate than random replacement. In addition, our method has a much lower modification rate (**19.7%**) compared to RockNER (40.2%), lower modification rate w.r.t original sentence indicates that the synthesized adversarial examples are semantically similar to the original sentence. After adversarial training, we observe that the model robustness improves and F1 under attack increases by **3.1%**.

---

### Official Review · Reviewer_RiQ4 · 2023-07-31

**Soundness:** 4

**Excitement:**

3: Ambivalent: It has merits (e.g., it reports state-of-the-art results, the idea is nice), but there are key weaknesses (e.g., it describes incremental work), and it can significantly benefit from another round of revision. However, I won't object to accepting it if my co-reviewers champion it.

**Paper Topic And Main Contributions:**

This paper focuses on improving the robustness of LLM NER models. To construct qualified adversarial examples, they propose an attack method incorporating disentanglement and word attribution methods. The disentanglement technique mitigates the bias between entity and non-entity words. For the substitution of entity and context words, they introduce a knowledge base and LLM to accomplish this process. Extensive experiments on three datasets show the effectiveness of their proposed method.

**Questions For The Authors:**

- What is the connection of the title and the main contributions in this paper?
- How does ChatGPT perform under the same attack perturbation?

**Reasons To Accept:**

- The research problem is interesting and practical.
- The motivation of mechanism design is clear.  Although the introduced mechanisms may have early existed, their applications are motivated by these findings. It makes those designs technical sound and easy to follow.
- Detailed case analysis and explanation.

**Reasons To Reject:**

- Threat model is not clear. The current manuscript lacks a subsection normally titled Threat Model, that describes the considering threat model that contains: 1) goals of the adversary; 2) capabilities of the adversary; 3) knowledge of the adversary); 4) goals of the defender; 5) capabilities of the defender; 6) knowledge of the defender.
- Formula (5) has an error and is missing an important item ${d\alpha}$
- The Context Word Substitution process is unclear. Please demonstrate the corresponding prompt or instruction.
- The abbreviation ``FOAK’’ in Line 100 should be accompanied by the full name. Moreover, there is no need to use this abbreviation, since I cannot find any other occurrence of FOAK in this paper.
- Lack of proof for validating the result of disentanglement.

**Reproducibility:**

4: Could mostly reproduce the results, but there may be some variation because of sample variance or minor variations in their interpretation of the protocol or method.

**Reviewer Confidence:**

4: Quite sure. I tried to check the important points carefully. It's unlikely, though conceivable, that I missed something that should affect my ratings.

---

> ### Author Rebuttal · Authors · 2023-08-29
>
> We thank the reviewer for their valuable inputs and feedback. We appreciate the reviewer recognizing the novelty of our method and exhaustiveness of the paper. Our response to your questions are as follows:
>
> **[Comparison with ChatGPT]:** While we do not have the result with ChatGPT, alternatively we now provide a comparison with **Llama2** here.
>
> | Adversarial Attack Results      | Attack Rate | Modification Rate |
> | ----------- | ----------- | ----------- |
> | Our Method    |  25.6%   |   19.7%   |
> | Baseline (RockNER)   |  26.7%      |  40.2%    |
> | Random  |  18.2%      |   19.7%    |
>
> In the table above, we report results using the Llama2-7b-chat-hf model on the CoNLL-2003 dataset. For this experiment, we allow a maximum of one word perturbation following our proposed framework. The first baseline method that we are comparing is RockNER, which by default substitutes all named entities as well as a maximum of three randomly selected context words. The second baseline method is the random replacement of a single word. Comparing the results of all three methods shown in the table, we observe that both our proposed method and RockNER achieve a higher attack rate than random replacement. In addition, our method has a much lower modification rate (**19.7%**) compared to RockNER (40.2%), lower modification rate w.r.t original sentence indicates that the synthesized adversarial examples are semantically similar to the original sentence. After adversarial training, we observe that the model robustness improves and F1 under attack increases by **3.1%**.
>
> **[Threat model subsection and Error in Eq (5)]:** In the final version of the paper we will add a subsection on the threat model. We thank the reviewer for pointing out the missing $d\alpha$ term in Eq (5) and we will address it in the final version.
>
> **[Context Word Substitution unclear]:** In the context word substitution, we mask the context word that needs to be replaced with [BLANK]. Then, we take the masked sentence as input to the LLM with a prompt such as “Please fill in the [BLANK] word with a single word, give me 5 possible candidate words: <MASKED SENT>”. Finally, we collect the word output as substitutions for the context word.
>
> **[Lack of proof validating disentanglement]:** Our framework is primarily driven by the intuition that applying disentanglement and attribution functions is a better approach for diversifying the adversarial example space hence assisting with adversarial robustness. The primary validation of this hypothesis is through the exhaustive empirical results and case study in the Appendix. We agree with the reviewer that theoretical analysis to study the limits of disentanglement and diversity in adversarial examples are good directions for future study.
>
> **[Acronym clarification]:** We can clarify in our writeup, but FOAK (First of a kind) is an existing public acronym which was not created by us in this paper. We will expand it in the final version and avoid usage of the acronym.
>
> **[Title and Contributions connection]:** We acknowledge the reviewer’s concern here, and we would like to clarify that the title was meant to highlight this completely unaddressed problem of adversarial robustness for LLM NER models rather than sounding as another proposed approach for an already existing problem. We believe the problem is of more importance which should be highlighted strongly and our solution is the first of a kind end-to-end pipeline to address this problem.

---

### Official Review · Reviewer_Yix7 · 2023-08-05

**Soundness:** 4

**Excitement:**

2: Mediocre: This paper makes marginal contributions (vs non-contemporaneous work), so I would rather not see it in the conference.

**Paper Topic And Main Contributions:**

This paper presents a novel attack that capitalizes on disentanglement and word attribution techniques, allowing for the learning of embeddings that capture both entity and non-entity influences separately. The latter approach is employed to identify key words that are important across both components. In contrast to most techniques, which primarily rely on manipulating non-entity words to generate adversarial examples, this approach expands the perturbation space to more effectively synthesize adversarial examples.


**Questions For The Authors:**


- In Figure 1, how do you perform PCA? Why do you think Rockner's results are worse? I can't tell which one is better.
  In the disentanglement procedure, do you use the initial word embeddings, or hidden states generated by the model?
- How do you define the context word? If the initial word embeddings are used, it would be rather strange since it has nothing to do with the model itself.
- It is interesting to see that you can attack T5's decoder by attacking a surrogate model: T5's encoder+ CRF. The instruction tuning and sequence labeling paradigms are quite different. Why can such an approach work so well?

**Reasons To Accept:**

- The paper proposes a word disentanglement approach to increase the performance of adversarial attacks.
- Result exceed previous baselines and the experiments are comprehensive.

**Reasons To Reject:**

- The motivation for the disentanglement is unclear. Related parts are set out in Appendix F, the very end of the manuscript. It should be highlighted and more quantitative and qualitative analysis is needed to support it. Why do context words affect the adversarial result, and why? Why disentangling such information may help the model produce better results? Since you have disentangled context information, is it possible to generate adversarial examples by simply substituting entity words? What is the upper bound of entity-word-only substitution? In my opinion, this is the most important contribution of this paper and needs further and deeper study.
- Some details are missing from the manuscript. See questions.
- I do not think this paper is related to LLM. BERT is not LLM. What is the model size of T5 you used? What is the result on modern LLMs, such as LLAMA?


**Reproducibility:**

4: Could mostly reproduce the results, but there may be some variation because of sample variance or minor variations in their interpretation of the protocol or method.

**Reviewer Confidence:**

4: Quite sure. I tried to check the important points carefully. It's unlikely, though conceivable, that I missed something that should affect my ratings.

---

> ### Author Rebuttal · Authors · 2023-08-29
>
> We thank the reviewer for their valuable inputs and feedback. We appreciate the reviewer for recognizing the effectiveness and comprehensive benchmarking of our proposed method. Our response to your questions are as follows:
>
> **[Comparison with Llama]:** We now provide a comparison with Llama2 also here.
>
> | Adversarial Attack Results      | Attack Rate | Modification Rate |
> | ----------- | ----------- | ----------- |
> | Our Method    |  25.6%   |   19.7%   |
> | Baseline (RockNER)   |  26.7%      |  40.2%    |
> | Random  |  18.2%      |   19.7%    |
>
> In the table above, we report results using the Llama2-7b-chat-hf model on the CoNLL-2003 dataset. For this experiment, we allow a maximum of one word perturbation following our proposed framework. The first baseline method that we are comparing is RockNER, which by default substitutes all named entities as well as a maximum of three randomly selected context words. The second baseline method is the random replacement of a single word. Comparing the results of all three methods shown in the table, we observe that both our proposed method and RockNER achieve a higher attack success rate than random replacement. In addition, our method has a much lower modification rate (**19.7%**) compared to RockNER (40.2%), lower modification rate w.r.t original sentence indicates that the synthesized adversarial examples are semantically similar to the original sentence. After adversarial training, we observe that the model robustness improves and F1 under attack increases by **3.1%**.
>
> **[Motivation for disentanglement]:** We appreciate the reviewer thoroughly reading our paper, and we can move contents from Appendix F to the main paper to provide more motivation on disentanglement in the main paper. The qualitative analysis part is interpreted primarily from our case study in Table 5 in the Appendix. After disentanglement, we rely on the word attribution function to sample entity and/or non-entity (context) components. We have done a separate study in Table 6 in the Appendix where we substitute only entity or non-entity components exclusively to estimate the upper bound, and we observe consistent patterns of much lowered F1 score compared to RockNER.
>
> **[Size of T5 model]:** The size of the T5 model used in this paper is the t5-large 770 million parameter model.
>
> **[RockNER comparison in Figure 1]:** Figure 1, is meant to highlight the fact that methods such as RockNER do not preserve the semantic similarity while doing perturbation while our method maintains semantic closeness to the original example without trading off on attack effectiveness. Further information on the generation of the embeddings visualized through PCA in Figure 1 is provided in the response below.
>
> **[T5 experiment setup and results]:** The left part of Figure 8 in the Appendix depicts how we train the initial T5 encoder in the Instruction NER setup and export this encoder to the figure on the right. Subsequently, we obtain embeddings from this T5 encoder which are then disentangled, and we then use the Integrated Gradients word attribution technique based on the T5 encoder + CRF model to identify important entity and non-entity (context) components. Recent separate work on (*One Embedder, Any Task: Instruction-Finetuned Text Embeddings, Su et al., ACL 2023 Findings*) has shown the utility of such instruct embeddings across other tasks and we believe there are interesting commonalities worth investigating further here as they don’t specifically evaluate adversarial attack effectiveness using instruct embeddings like we do.

---

### Official Review · Reviewer_fP4e · 2023-08-12

**Soundness:** 3

**Excitement:**

2: Mediocre: This paper makes marginal contributions (vs non-contemporaneous work), so I would rather not see it in the conference.

**Paper Topic And Main Contributions:**

The paper addresses the limitations of Large Language Models (LLMs) in complex Named Entity Recognition (NER) tasks. In particular, it aims at improving the LLM robustness, introducing a pipeline mainly based on a novel adversarial attack disentangling entity and non-entity representations via an autoencoder approach to reduce the bias in the selection of words. This experimental assessment shows an improvement in terms of the F1 score of LLM NER models on datasets such as CoNLL-2003 and Ontonotes 5.0.

**Reasons To Accept:**

The problem addressed is of interest to the research community. The approach is overall sounding and combines existing techniques in the literature (e.g., disentanglement via autoencoders, sentence embedding, and word attribution) in a simple and reasonable pipeline.

**Reasons To Reject:**

The novelty of the paper is rather limited, the analyses of the disentangling mechanism are rather high-level (it is a given, never really analysed, what kind of information would be preserved once the representations of the entities are disentangled from the context, everything is justified only by the final performance on two datasets).
The experimental analyses, while written fluently, is difficult to follow and the language is at time verbose.

Secondary:
The overall paper is well-written in English, but there are often logical gaps in the structure of the paragraphs.
Just to mention a few in the Introduction, but several are spread throughout the paper:
Line 042: There is a long introduction to LLM and their hallucination, without mentioning any possible link of this problem to the NER tasks;
Line 071: it introduces the concept of "steps", but no "process" was so far mentioned;
Line 193-195: it introduces "for the sake of brevity" abbreviations that are rather obscure and probably do not actually abbreviate much the text;
Line 232: mentioned "3 neural networks" while this architecture is then discussed as an end-to-end architecture, without particular discontinuity (please refer to the literature about AE and VAE).
Similar problems made the reading of the experimental evaluation difficult to assess.

**Reproducibility:**

4: Could mostly reproduce the results, but there may be some variation because of sample variance or minor variations in their interpretation of the protocol or method.

**Reviewer Confidence:**

3: Pretty sure, but there's a chance I missed something. Although I have a good feel for this area in general, I did not carefully check the paper's details, e.g., the math, experimental design, or novelty.

**Typos Grammar Style And Presentation Improvements:**

I would invite the authors to rephrase the paragraphs to be less verbose to improve the overall readability of the paper.
Additional problems:
Please fix references to Figure following EMNLP former (e.g., Line 068)
Please fix references to work directly discussed (e.g., line 168)
Line 058: remove contractions
Line 078: remove "an"

---

> ### Author Rebuttal · Authors · 2023-08-29
>
> We thank the reviewer for their valuable inputs and feedback. We appreciate the reviewer for recognizing the soundness and effectiveness of our proposed method. Our response to your questions are as follows:
>
> **[Clarifications in the writeup]**: In Line 42, the LLM hallucination problem is mentioned as it is attributed as one of the primary reasons for the poor robustness of LLM NER’s (GPT-NER paper referenced on Line 761) which is relevant to the problem we are addressing in our paper. In Line 71, we wanted to guide the reader to look at Figure 2 which outlines the overall process flow while building a high level overview of the steps in the narrative. In Line 232, the three neural networks mentioned are an intrinsic part of the disentanglement module which fits within the bigger end-to-end architecture proposed in our paper. We will work towards fixing the references and making these subsections as well as parts of the experimental setup less verbose to improve the readability of the final version.
>
> **[Analysis of disentanglement]**: We agree with the reviewer that the primary validation of our hypothesis on the effectiveness of disentanglement and word attributions is obtained through the benchmarking done on the three datasets. However, beyond empirical validation, we also provide visual insights on how disentanglement modifies the representation through Figure 2. Eq (4) ensures that the output entity representation ($\hat{E}$) after disentanglement in Figure 2 is preserving the necessary information from the original entity representation ($E$). Additional qualitative insights from the case study in Table 5 in the Appendix also provide more insight into the sentences generated post disentanglement representing preserved and modified components explicitly. We will consolidate this analysis even further as we expand upon this work.

---

### Meta-Review · Area_Chair_L3Qg · 2023-09-17

**Recommendation:** 3

**Metareview:**

This paper presents a method to generate adversarial examples based on disentanglement and word attribution techniques. Extensive experiments show the effectiveness of the proposed method. Based on the initial reviews, the reviewers found this paper working on an interesting problem, proposing a reasonable method, and conducting comprehensive experiments. However, there are many important details missing regarding, for example, the threat model, the context word substitution process, the motivation and analysis of disentanglement. Also, there is a question raised whether the tested models (i.e., BERT-base and T5-large) are really LLMs.

In the rebuttal, the authors addressed most of the reviewers' concerns and answered detailed questions. Despite presenting additional results on Llama, the paper would be more interesting if the authors focused specifically on large language models as the NER robustness of pre-trained models has been studied in the literature while we know much less about how the emergent ability of LLMs (due to their enormous size) responds to the adversarial examples. Also, based on the threat model explained, the adversary works on the white-box scenario (with access to NER model's architecture and training data) of which the specific application value is not quite clear.

If the paper is accepted, please consider fusing the rebuttals into the camera-ready version (especially on the threat model, the analysis of disentanglement, and the results on Llama) and revising the title (as discussed) so it better reflects the actual contributions of the paper.

---

### Decision · Program_Chairs · 2023-10-07

**Decision:**

Accept-Findings

**Comment:**

This paper presents a method to generate adversarial examples based on disentanglement and word attribution techniques. Extensive experiments show the effectiveness of the proposed method. Based on the initial reviews, the reviewers found this paper working on an interesting problem, proposing a reasonable method, and conducting comprehensive experiments. However, there are many important details missing regarding, for example, the threat model, the context word substitution process, the motivation and analysis of disentanglement. Also, there is a question raised whether the tested models (i.e., BERT-base and T5-large) are really LLMs.

In the rebuttal, the authors addressed most of the reviewers' concerns and answered detailed questions. Despite presenting additional results on Llama, the paper would be more interesting if the authors focused specifically on large language models as the NER robustness of pre-trained models has been studied in the literature while we know much less about how the emergent ability of LLMs (due to their enormous size) responds to the adversarial examples. Also, based on the threat model explained, the adversary works on the white-box scenario (with access to NER model's architecture and training data) of which the specific application value is not quite clear.

If the paper is accepted, please consider fusing the rebuttals into the camera-ready version (especially on the threat model, the analysis of disentanglement, and the results on Llama) and revising the title (as discussed) so it better reflects the actual contributions of the paper.